# LARGE LANGUAGE MODELS AS IMPROVEMENT OPERATORS: BETTER REASONING BY ITERATION

## ABSTRACT

Reasoning training of LLMs teaches them to produce long chains of thought (long CoT), with attendant increase in accuracy (good!), context length (bad!), compute cost (bad!) and answer latency (bad!). Can current models provide other combinations on this Pareto frontier, e.g., give answers with better accuracy than long CoT models, despite lower context length and/or latency? We show that the answer is "yes" and also give ways to systematically think about these design choices. Abstractly, this involves viewing the model as an *improvement operator* with a continuum of strategies for improving its problem solving, (example: generate four shorter answers and combine their good points in a single superior answer). We study an inference method **Parallel-Distill-Refine (PDR)** that performs a few rounds of the following: (i) generate diverse drafts in parallel; (ii) *distill* them into a bounded, textual *workspace*; and (iii) *refine* conditioned on this workspace, which then seeds the next round. PDR often provides better performance than long CoT and has lower latency and context size. An interesting subcase of PDR is **Sequential Refinement (SR)**, which iteratively improves a single candidate answer without a persistent workspace. It provides performance superior to long CoT, with the benefit of compact context size but high latency. These examples suggest training interventions to shift the Pareto frontier. For example, we use RL to improve an $8B$ model to better align with **PDR** as the inference method, which improves performance. On math tasks with rule-based checkers, iterative pipelines surpass single-pass baselines at matched sequential budget; shallow **PDR** delivers the largest gains (e.g., +10% on AIME 2024 and +11% on AIME 2025).

## 1 INTRODUCTION

Scaling language models to solve harder problems has increasingly relied on eliciting explicit reasoning traces ("thinking tokens") at inference time (Wei et al., 2022; Jaech et al., 2024; Guo et al., 2025). While longer traces often correlate with accuracy, they entangle reasoning depth with sequence length and inherit long-context failure modes (Ghosal et al., 2025). In parallel, the field is gravitating toward *self-improvement*: systems that refine their own outputs via critique, revision, debate, or sample-and-select without expert supervision (Gou et al., 2023; Du et al., 2023b; Irving et al., 2018; Yao et al., 2023; Pan et al., 2025; Zhang et al., 2025).

Stepping back from the rich body of work, one begins to see LLM inference as a malleable concept: instead of a single "reasoning trace" one faces a menu of choices from some primitives: generate fresh answers; critique/revise/debate/summarize generated answers; create updated answer. With this choice comes an unexplored Pareto frontier:

> *What is the best possible task accuracy achievable after fixing desired budget constraints, e.g.: (i) total tokens across all generations (ii) max depth of the generation chain (roughly, "latency") (iii) total context length (iv) total compute (which depends on all of the above in a complicated way).*

The confounding factor is that iteration alone does not guarantee progress. Simply asking the model to "try again" risks forgetting useful partial results and repeating earlier mistakes. Naïvely appending all prior attempts to the context recreates long-context failures and scales cost with the number of rounds. Models suffer from (possibly due to the nature of the attention mechanism) anchoring

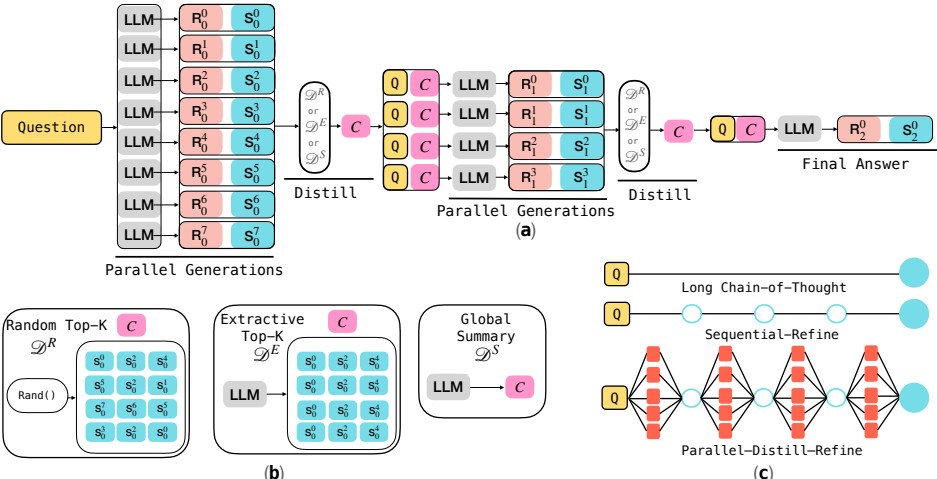

Figure 1: (a) **Parallel-Distill-Refine (PDR)** instantiation. Each round $r$ generates $M_r$ parallel drafts which are then distilled into the workspace following one of the distillation approaches illustrated in (b). (c) Three inference regimes. Top: Long chain-of-thought with a single, long trace. Middle: Sequential Refinement (**SR**) where a single draft is updated over short rounds. Bottom: Parallel-Distill-Refine (**PDR**) where each round spawns $K$ parallel drafts, then distills and refines to one updated draft. All panels keep the per-call sequential budget $B_{\text{seq}}$ fixed; **PDR** improves accuracy by increasing total compute $B_{\text{total}}$ via parallelism while maintaining similar latency.

biases (see Figure 4, 6) as well as forgetfulness. A viable scheme needs a compact state that (i) carries forward salient facts and intermediate results, (ii) flags disagreements and open subgoals, and (iii) remains bounded so each generation (and overall context-size) stays short.

This paper studies inference strategies that generate many tokens with a compact context size. Instead of long chains of thought, inference has phases that generate solutions within the allowed token budget and then write a bounded, round-wise summary/report (e.g., listing agreements, contradictions, intermediate results, and open subgoals). The next phase starts with only this summary and uses available workspace for fresh generations (which benefit from accumulated wisdom in the summary). Iterating this process can generate long "thinking" albeit in bounded context size. [1]

We study two instantiations: (i) *Sequential refinement (SR)* (no workspace), where a single artifact (solution, proof, program) is iteratively improved for a fixed number of steps; and (ii) *Parallel-Distill-Refine (PDR)* (round-wise workspace), where each round samples $M$ drafts in parallel, distills them into a bounded summary for the next round, and continues. The workspace is not persistent across rounds; it is freshly synthesized for each round. A central challenge is information synthesis: compressing salient facts and intermediate results; flagging uncertainty and unresolved subgoals; and retiring stale or contradicted notes. Its effectiveness hinges on four meta-skills: *verification* (detect and localize errors via self-judging and cross-candidate checks), *refinement* (use feedback-/context to improve the artifact), *compression* (retain only salient history via bounded summaries rather than replay), and *diversification* (sustain exploratory variation to avoid consensus collapse).

**Learning to improve short-context iteration.** It is also of interest to teach the model a policy that effectively leverages this improvement operator. Standard RL for training reasoning models typically optimizes a single, long chain-of-thought conditioned on the prompt, with reward on the final answer. **PDR**, by contrast, comprises multiple short iterations that read a bounded summary, write a refinement, and re-synthesize a fresh summary. This creates a train–test mismatch in the information flow (short updates vs. one long trace). To align training with deployment, we optimize an objective that unrolls the operator itself during training: sample $M$ short drafts, distill them into a compact bounded summary, and condition on the prompt plus that summary to produce a

---

[1]Such LLMs fall within a traditional framework of *randomized space-bounded computation*. Computational complexity theory Arora & Barak (2007) shows it is capable of surprisingly powerful reasoning, such as determining connectivity of much larger graphs that cannot even fit the working memory; see Section B.

refined attempt; verifiable rewards supervise the end-to-end computation. This objective narrows the train–test gap while holding sequential latency fixed.

**Results.** On math tasks with rule-based checkers, iterative pipelines surpass single-pass baselines at matched sequential budget; shallow `PDR` delivers the largest gains (e.g., +10% on AIME 2024 and +11% on AIME 2025). Beyond orchestration, aligning training with inference, via an operator-consistent RL objective that optimizes the same read/write interface used at test time yields further improvements (e.g., $\sim 5\%$ on AIME 2024 and AIME 2025 when mixing standard and operator RL). To evaluate latency and compute we report accuracy under two budgets: a *sequential budget* (tokens along the accepted path, a latency proxy) and a *total budget* (all tokens spent across calls, a cost proxy). These findings suggest that iteration with short contexts and compact summaries can substitute for long traces while holding latency fixed.

## 2 LLMs as Improvement Operators

### 2.1 Problem setting and notation

We consider tasks $x$ (e.g., a math problem) and aim to produce a high-quality final artifact $s_{\text{final}}$ (solution, proof, or program) under a given token budget. Let $\mathcal{M}_\theta$ denote a (frozen or trainable) LLM used as an *improvement operator*. Given a current artifact $s_t$ and a compact textual workspace $C_t$, the model proposes a refinement:

$$s_{t+1} \leftarrow \mathcal{M}_\theta(x, s_t, C_t). \tag{1}$$

The workspace $C_t$ is a bounded summary ($|C_t| \leq \kappa$ tokens) meant to capture agreements, contradictions, intermediate results, and open subgoals.

**Read-write-compress cycle.** Each step (i) reads the current workspace $C_t$, (ii) writes a refined artifact $s_{t+1}$ via $\mathcal{M}_\theta$, and (iii) compresses back into a bounded workspace for the next step using a synthesis operator $\mathcal{D}$:

$$C_{t+1} \leftarrow \mathcal{D}(x, s_{t+1}), \qquad |C_{t+1}| \leq \kappa. \tag{2}$$

**Token budgets.** We evaluate every method under two budgets:

$$B_{\text{seq}} = \sum_{c \in \mathcal{P}} \left( \text{in}_c + \text{out}_c \right) \qquad \text{(latency proxy; tokens along the accepted path),} \tag{3}$$

$$B_{\text{total}} = \sum_{c=1}^{C} \left( \text{in}_c + \text{out}_c \right) \quad \text{(compute/cost proxy; all calls, including discarded branches).} \tag{4}$$

Here $c = 1, \dots, C$ indexes all model calls (prompts, candidate generations, and distillation/summary updates); $\text{in}_c$ and $\text{out}_c$ are the input and output tokens for call $c$; and $\mathcal{P} \subseteq \{1, \dots, C\}$ is the final accepted path. We report accuracy as a function of both axes and match baselines per axis (e.g., equal $B_{\text{seq}}$ for latency-controlled comparisons).

### 2.2 Instantiations

We study two short-context iterative refinement pipelines.

#### 2.2.1 Sequential refinement (`SR`; depth over a single candidate).

We set $C_t \equiv \varnothing$ for all $t$ and iteratively improve a single artifact for $R$ rounds:

$$s_{t+1} \leftarrow \mathcal{M}_\theta(x, s_t, \varnothing), \quad t = 0, \dots, R-1, \qquad s_{\text{final}} = s_R. \tag{5}$$

### 2.2.2 PARALLEL-DISTILL-REFINE (**PDR**; ROUND-WISE WORKSPACE)

We do not maintain a persistent memory. Instead, for rounds $r = 1, \ldots, R$; we sample $M_r$ drafts (Parallel) conditioned on the current bounded summary, then re-synthesize (Distill) a fresh bounded summary for the next round:

$$\text{(Parallel)} \quad S^{(r)} = \big\{ s_i^{(r)} \leftarrow \mathcal{M}_\theta(x, C^{(r-1)}) \big\}_{i=1}^{M_r}, \qquad C^{(0)} = \varnothing, \tag{6}$$

$$\text{(Distill)} \quad C^{(r)} \leftarrow \mathcal{D}\big(x, S^{(r)}\big), \qquad |C^{(r)}| \leq \kappa. \tag{7}$$

We enforce a last-round fan-out of $M_R = 1$; the single generation in round $R$ is returned as $s_{\text{final}}$. The summary is round-wise and non-persistent: earlier text is not replayed, preventing growth in per-call context.

**Why a round-wise summary?** Replay of all prior attempts scales linearly with steps and reintroduces long-context failure modes. Re-synthesizing $C^{(r)}$ from the current drafts keeps the memory *bounded* ($|C^{(r)}| \leq \kappa$) and focuses each round on the most recent and informative evidence.

**Constructing the compact summary** $C^{(r)}$**.** We consider several practical instantiations of the distillation operator $\mathcal{D}$, all obeying $|C^{(r)}| \leq \kappa$:

- **Global summary:** Produce a single shared $C^{(r)}$ that captures agreements, contradictions, derived facts, unresolved subgoals, and next actions. This emphasizes verification and comparison while retiring stale or contradicted notes.

- **Extractive top-$k$ evidence (shared).** Instead of free-form text, select the $k$ solutions from $S^{(r)}$ as the workspace itself, trading compression for higher fidelity to the best evidence.

- **Random-$k$ / bootstrapped workspaces.** For the next round, construct multiple small workspaces by randomly sampling $k$ solutions per generation. This injects diversity and mitigates premature consensus while keeping each workspace small.

**Budgets.** Tokens used for **Parallel**, **Distill**, and **Refine** contribute to $B_{\text{total}}$. The reported latency $B_{\text{seq}}$ counts only the tokens on the accepted generate→distill→refine path for the final output.

## 2.3 OPERATOR-CONSISTENT TRAINING

The previous sections treat $\mathcal{M}_\theta$ as frozen and rely purely on prompting/orchestration. We now align training with deployment/inference by optimizing the model under the same short-context, iterative interface used at test time.

**Motivation.** Most RL for reasoning LLMs optimizes a single, long chain-of-thought trajectory. If inference instead runs multiple short passes with a compact workspace $C$, this creates a train-test mismatch. We remove this mismatch by mixing two training modes: (i) standard long-trace optimization, and (ii) *operator rollouts* that execute the generate→distill→refine interface under short contexts.

**Base Algorithm.** For the baseline RL, we use the CISPO objective from Minimax-M1 (Li et al., 2025). For a given prompt $x$, the generator $\pi(\cdot \mid \theta_{\text{old}})$ generates $G$ rollouts $\{o_{i=1}^G\}$ using the old policy $\theta_{\text{old}}$. Automated checkers like sympy (Meurer et al., 2017) or math-verify[2] are used to assign scalar rewards $r_i$ ($\pm 1$) to each of the rollouts. CISPO combines the group-normalized advantage from GRPO (Shao et al., 2024) with REINFORCE (Williams, 1992) to achieve the following objective:

$$\mathcal{J}_{\text{CISPO}}(\theta) = \mathbb{E}_{x \sim \mathcal{D}, \{o_i\}_{i=1}^G \sim \pi(\cdot|x,\theta_{\text{old}})} \left[ \frac{1}{\sum_{i=1}^G |o_i|} \sum_{i=1}^G \sum_{t=1}^{|o_i|} \text{sg}(r_{i,t}(\theta)) A_i \log(\pi_\theta(o_{i,t}|x, o_{i,<t})) \right] \tag{8}$$

where $A_i = \frac{r_i - \text{mean}(\{r\}_{j=1}^G)}{\text{std}(\{r\}_{j=1}^G)}$ is the advantage, sg is the stop-gradient operation, and $r_{i,t}(\theta)$ is computed using the asymmetric clipping from Yu et al. (2025) as follows:

---

[2]https://github.com/huggingface/Math-Verify

$$r_{i,t} = \text{clip}\left(\frac{\pi_\theta(o_i \mid x, o_{i,<t})}{\pi_{\theta_{\text{old}}}(o_i \mid x, o_{i,<t})}, 1 - \epsilon^-, 1 + \epsilon^+\right) \tag{9}$$

where $\frac{\pi_\theta(o_i|x,o_{i,<t})}{\pi_{\theta_{\text{old}}}(o_i|x,o_{i,<t})}$ is the importance-sampling (IS) weight. Additionally, we add an SFT loss (negative log-likelihood) similar to Seed et al. (2025) on rollouts which lead to positive rewards. The final training objective becomes:

$$\mathcal{J}(\theta) = \mathcal{J}_{\text{CISPO}}(\theta) + \alpha \cdot \mathcal{J}_{\text{SFT}}(\theta) \tag{10}$$

where $\alpha$ is usually set to a small value like $0.1$. The addition of this SFT loss boosts the utilization of positive rollouts and enforces better verification behavior in model training.

**Data mixture.** At each update, draw a mini-batch $\mathcal{B} = \{x_i\}_{i=1}^N$ and split it evenly into two sub-batches $\mathcal{B}_{\text{trace}}$ and $\mathcal{B}_{\text{op}}$ with $|\mathcal{B}_{\text{trace}}| = \lfloor N/2 \rfloor$ and $|\mathcal{B}_{\text{op}}| = \lceil N/2 \rceil$. We train on $\mathcal{B}_{\text{trace}}$ with a standard long-trace objective $\mathcal{J}_{\text{trace}}(\theta)$, and on $\mathcal{B}_{\text{op}}$ with *operator rollouts* under short context, yielding $\mathcal{J}_{\text{op}}(\theta)$. The per-step objective is the average of the two:

$$\mathcal{J}_{\text{train}}(\theta) = \tfrac{1}{2}\mathcal{J}_{\text{trace}}^{\mathcal{B}_{\text{trace}}}(\theta) + \tfrac{1}{2}\mathcal{J}_{\text{op}}^{\mathcal{B}_{\text{op}}}(\theta), \tag{11}$$

where $\mathcal{J}_{\text{trace}}^{\mathcal{B}_{\text{trace}}}$ and $\mathcal{J}_{\text{op}}^{\mathcal{B}_{\text{op}}}$ denote the losses computed on their respective sub-batches. Other ratios are possible; we use a 1:1 split in our experiments.

**Mode A: Standard long-trace optimization.** Given $x$, sample a single, long trajectory $s_{1:T} \sim \mathcal{M}_\theta(x)$ and optimize a conventional RL verifiable signal (e.g., a rule based verifiable reward for math problems). This preserves the model's ability to reason in extended traces when available.

**Mode B: Operator rollouts under short context.** We roll out the same interface used at test time but with one round for stability and cost.

*(i) Parallel-Distill-Refine (**PDR**; one-round rollout).*

1. Generate $M$ parallel generations (reasoning traces, solutions) conditioned on an empty summary:
$$S = \{s_i \leftarrow \mathcal{M}_\theta(x, C^{(0)}=\varnothing)\}_{i=1}^M.$$

2. Distill to a bounded, round-wise summary $C$.
3. Refine a single candidate conditioned on $C$: $\tilde{s} \leftarrow \mathcal{M}_\theta(x, s_j, C)$.

**Why one round during training?** Rolling out a single **PDR** round (with $M$ early drafts, distillation to $C$, and a single refinement) captures the key interface while controlling $B_{\text{total}}$ and stabilizing RL. At inference we can run multiple rounds ($R > 1$) using the same operator.

Our datamix preserves competence on long traces while teaching the model to reason across short iterations. **PDR** is emulated by a one-round of parallel→distill→refine rollout where the model observes $(x, C)$ and is optimized with a verifiable reward on the final solution trace.

## 3 EXPERIMENTS

In this section we present evidence for the central claim of the paper that short-context iteration with a bounded, round-wise workspace can substitute for long chains of thought while controlling latency and total compute. We compare the Sequential refinement (**SR**) and Parallel-Distill-Refine (**PDR**) operators against long chain-of-thought baselines under a budget-aware protocol. Unless stated otherwise, we measure accuracy with rule-based verifiers like sympy (Meurer et al., 2017). Additionally, we report the results as functions of both the sequential budget $B_{\text{seq}}$ (latency proxy along the accepted path) and the total budget $B_{\text{total}}$ (all tokens across calls).

We try to answer the following four research questions through our experiments:

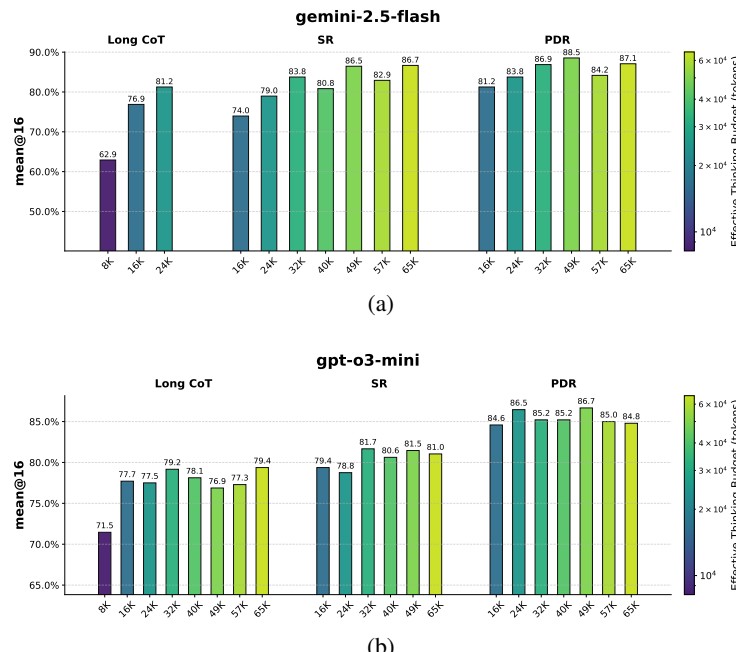

Figure 2: **AIME 2024: Iterative improvement beats single-pass long-CoT at matched sequential budgets.** The $x$-axis reports $B_{\text{seq}}$: the thinking tokens consumed along the accepted path of the iterative chain, plus any distilled summary that conditions the next step. Tokens spent on unused parallel proposals are excluded, so $B_{\text{seq}}$ serves as a latency proxy. At comparable $B_{\text{seq}}$, both **SR** and **PDR** outperform the single-pass long CoT baseline, with **PDR** yielding the largest gains by converting additional total compute (via parallelism) into accuracy without increasing per-call context.

- **RQ1:** Can short-context iterations beat long traces at matched latency by comparing $\{$**SR** , **PDR**$\}$ to long-trace CoT at matched $B_{\text{seq}}$ and $B_{\text{total}}$.

- **RQ2:** Figuring out the best distillation strategy for producing $C^{(r)}$ by comparing three $\mathcal{D}$ variants: global summary, extractive top-$k$, and random-$k$ bootstraps.

- **RQ3:** Identifying the effect of the verification ability of a given model on the final performance.

- **RQ4:** Whether operator-consistent training (i.e., make it better at what it needs to do at test time) shifts the Pareto Frontier. This will compare a operator-consistent + standard RL with standard single-trace RL (Sec. 2.3) at matched $B_{\text{seq}}$.

**Setup.** We evaluate **SR** and **PDR** as inference-time operators on math problems. Given a prompt $x$, the model produces a thinking trace and a final solution. The thinking spans, delimited by `<think>` ... `</think>` are stripped out and only the self-contained solutions are used to build the inputs for subsequent rounds. We evaluate on AIME 2024 and AIME 2025 (AoPS, 2025) and report accuracy computed using math-verify[3].

**Models and budgets.** The models considered are `o3-mini` ("medium" reasoning effort) (OpenAI, 2025) and `gemini-2.5-flash` (Comanici et al., 2025). For `gemini-2.5-flash`, we vary the thinking budget from 8192 to 24576 tokens (the maximum supported), while reserving an additional 2048 tokens for the final solution. `o3-mini` does not expose a separate thinking budget, therefore we vary the maximum generation length from 10240 to 26624, which corresponds to the same total allowance (assuming 8192-24576 tokens for thinking plus an additional 2048 tokens for the solution). Both operators are run under matched sequential budgets while allowing different total token budgets via parallelism. We run inference using temperature and `top-p` values of 1.0.

**RQ1: Do short-context iterations beat long traces at matched latency?**

---

[3]https://github.com/huggingface/Math-Verify

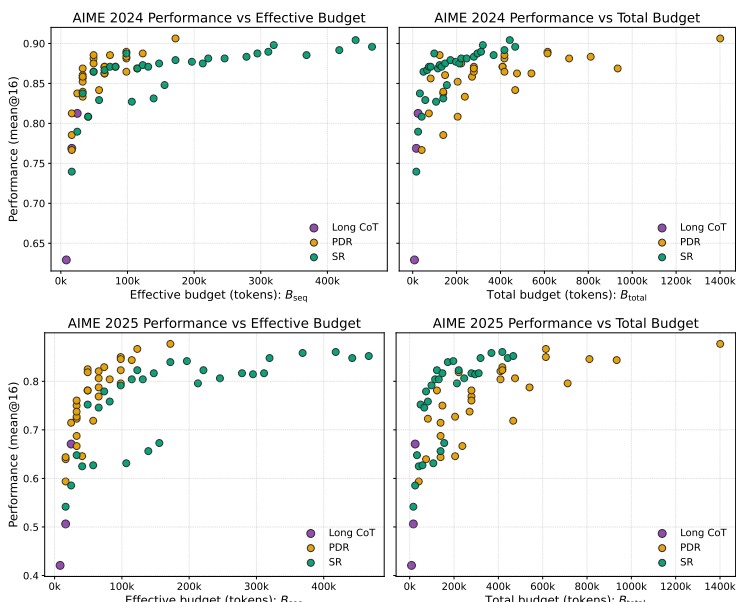

Figure 3: **Token Budgets comparison:** We plot all the different configurations for Long CoT, **SR** and **PDR** operators for both $B_{\text{seq}}$ and $B_{\text{total}}$ token budgets for gemini-2.5-flash. For $B_{\text{seq}}$, **PDR** forms the Pareto-frontier and gives consistent gains over Long CoT and **SR**. However, for $B_{\text{total}}$, **SR** forms the pareto-frontier because there are no parallel drafts involved so no generations are discarded.

**Sequential Refinement (SR).** For the **SR** operator, we run o3-mini and gemini-2.5-flash for thinking budgets $B \in \{8192, 16384, 24576\}$ and refinement rounds $r \in \{1, \ldots, 6\}$. The prompt template is given in §A.1.1.

**Parallel-Distill-Refine (PDR).** We evaluate **PDR** under a fixed thinking budget $B$ using three settings. These settings differ by number of rounds, number of parallel generations in each round and selecting the $k$ candidate solutions to carry forward via textual workspace : $g = [8]$, $k$=4; $g = [16, 8]$, $k$=4; and $g = [16, 8, 4]$, $k$=2. Here $g = [g_1, \ldots, g_r]$ specifies the number of parallel generations in each round, and $k \leq \min_d g_r$ is the number of candidates forwarded to the next round. For distillation (i.e., selecting the $k$ candidates to carry forward), we compare: Random-$k$ (uniform sampling per instance); Top-$k$ (model-graded) where the same base model assigns a quality score to each candidate and we keep the top $k$ per instance (we report both a shared rubric and a per-instance grading prompt); and global-summary that aggregates all candidates using a summarization prompt. Refinement, selection, and summarization prompts are detailed in §A.1.2.

**Results.** Figures 2 and 7 report accuracy on AIME 2024 and AIME 2025 under matched effective token budgets (same total tokens, including thinking and final answer). We observe consistent gains when moving from long chain-of-thought to **SR**, which continue when moving from **SR** to **PDR**. For o3-mini at an effective budget of 49k tokens with a per-call thinking budget of 16k, accuracy improves from 76.9 (Long CoT) to 81.5 (**SR**) and 86.7 (**PDR**), an absolute improvement of $+9.8$ percentage points over Long CoT. gemini-2.5-flash shows smaller deltas from **SR** to **PDR** than o3-mini, suggesting stronger intrinsic self-verification in gemini-2.5-flash. AIME 2025 exhibits similar trends.

**RQ2: Which distillation (i.e., summarization) strategy works best?**

Table 1 studies the distillation operator $\mathcal{D}$ in **PDR** under a (fixed number of rounds, number of generations in each round) setting $g = [16, 8, 4]$ with $k = 2$ candidates per round. Across datasets and base models, *per-sample top-$k$* and *global-summary* selection consistently outperform *shared top-$k$* and *random-$k$*, and the margin widens as the thinking budget $B$ increases. The main exception is AIME 2025 with o3-mini, where *global summary* outperforms the alternatives. We hypothesize

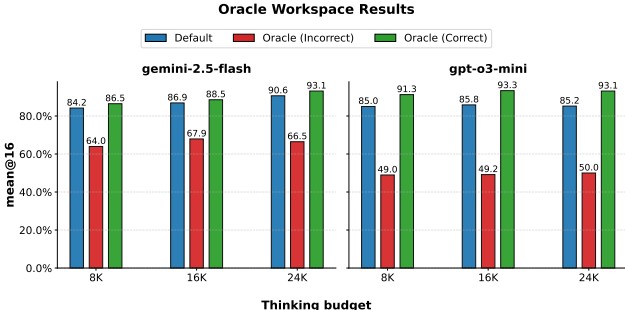

Figure 4: **AIME 2024: Anchoring bias due to +ve and −ve examples:** With **PDR** we compare three selection policies for the summary: Random-$k$, Oracle-Incorrect (all $k$ candidates are incorrect), and Oracle-Correct (all $k$ candidates are correct), evaluated on both `gemini-2.5-flash` and `o3-mini`. Across all thinking budgets, admitting only incorrect candidates into the summary yields a pronounced drop in accuracy, whereas admitting only correct candidates improves over the Random-$k$ baseline. The degradation under Oracle-Incorrect is markedly larger for `o3-mini` than for `gemini-2.5-flash`, indicating weaker self-verification in `o3-mini`.

Table 1: **Effect of distillation operator** $\mathcal{D}$**:** We compare the effect on final performance by changing the distillation operator $\mathcal{D}$. Each table column reports accuracies on AIME 2024 / AIME 2025. We compare four choices: (i) *Global summary*: aggregate all candidates and synthesizes a single compact summary; (ii) *Per-sample top-$k$*: each downstream branch selects its own top-$k$ candidates as the summary; (iii) *Shared top-$k$*: a single set of top-$k$ candidates is shared as the summary across generations for next round; (iv) *Random-$k$*: each generation for next round receives $k$ candidates sampled uniformly at random for the summary. Overall, global summary and per-sample top-$k$ tend to perform best, with gains more pronounced at higher thinking budgets. For o3-mini on AIME 2025, global summary yields the largest improvement, suggesting strong summarization ability in o3-mini. We use $k = 2$ for these experiments.

| Budget | gemini-2.5-flash | | | | gpt-o3-mini | | | |
|---|---|---|---|---|---|---|---|---|
| | Global | PS top-$k$ | Shared top-$k$ | Random-$k$ | Global | PS top-$k$ | Shared top-$k$ | Random-$k$ |
| 8192 | 83.13 / 66.88 | 83.75 / 71.88 | 84.17 / 70.21 | 83.33 / 66.67 | 86.04 / 82.92 | 84.79 / 76.04 | 85.00 / 76.67 | 82.50 / 71.25 |
| 16384 | 86.46 / 84.38 | 86.88 / 83.96 | 86.46 / 83.75 | 86.25 / 80.63 | 86.46 / 84.79 | 85.42 / 74.58 | 85.83 / 76.88 | 83.13 / 71.88 |
| 24576 | 88.75 / 87.71 | 90.63 / 85.00 | 87.71 / 85.42 | 88.13 / 79.58 | 85.42 / 83.54 | 85.21 / 77.92 | 85.00 / 75.42 | 82.29 / 72.08 |

that `o3-mini`'s summarization is particularly effective at capturing cues from both correct and incorrect drafts, and these cues, when distilled, lead to stronger subsequent refinements.

**RQ3: Effect of model verification abilities.**

**Oracle PDR analysis.** To understand the role of model verification within **PDR**, we intervene on the set of candidates admitted to the summary at each round. We use a three-round **PDR** with number of generations in each round as $[16, 8, 4]$ and top-$k$ selection ($k = 2$), and compare: (i) **Random-$k$:** choose $k$ candidates uniformly at random from the previous depth; (ii) **Oracle (Correct):** admit only correct candidates to the compact summary when available; (iii) **Oracle (Incorrect):** admit only incorrect candidates.

Firstly, from Figures 4 and 6, we can see that injecting incorrect candidates (Oracle (Incorrect)) causes large drops for all models. The degradation is substantially larger for `o3-mini` than for `gemini-2.5-flash`, suggesting stronger self-verification and recovery in the latter. The same trend holds across AIME 2024 and AIME 2025.

**RQ4: Does operator-consistent training move the Pareto Frontier?**

**Training setup.** We train an 8B dense model similar to Llama-3-style architecture (Dubey et al., 2024). For warm-start supervised fine-tuning (SFT), we use GPT-OSS 120B (Agarwal et al., 2025) to generate synthetic traces for math and code prompts sampled from Polaris-53K (An et al., 2025) and DeepCoder (Luo et al., 2025), respectively. We run SFT for 8B tokens (∼4 epochs). For RL training, we use the Polaris-53K dataset. Both SFT and RL datasets are decontaminated against

Table 2: **Operator RL results:** Comparison of RL training objectives on AIME 2024/2025 at matched sequential budget $B_{\text{seq}} = 65{,}536$ using a dense 8B model. Mixing standard RL with operator-consistent RL (Op-RL) yields consistent gains for iterative inference operators such as **PDR** while preserving performance on the Long CoT baseline. Op-RL can also be applied as a continual RL to existing baseline RL checkpoint.

| Model | AIME 2024 | | AIME 2025 | |
|---|---|---|---|---|
| | Long CoT | **PDR** | Long CoT | **PDR** |
| 8B SFT Policy | 47.50 | 62.92 | 35.00 | 47.50 |
| 8B Baseline RL | 67.50 | 75.83 | 59.58 | 65.83 |
| 8B **PDR** RL | 69.58 | 79.17 | 57.50 | 67.50 |
| 8B Continual **PDR** RL | 70.00 | 80.83 | 61.25 | 70.42 |

AIME 2024/2025 (AoPS, 2025) and MATH-500 (Hendrycks et al., 2021). Further details and hyperparameters are detailed in Appendix A.2.

**Baseline RL** As described in Section 2.3, we use the CISPO objective for RL post-training (Li et al., 2025). We set $\epsilon^- = 0.0$ and $\epsilon^+ = 5.0$, and remove "zero-variance" prompts from a given batch (Seed et al., 2025). We use forced interruptions (Hong et al., 2025; Yang et al., 2025) to control generation length from exploding by injecting the phrase "`Okay, time is up. Let me stop thinking and formulate a final answer now.</think>`" after a thinking budget of $16{,}384$ tokens. We additionally keep a buffer of 2048 tokens for the final solution, thus keeping a maximum generation length of 18432. 32 generations are sampled per prompt with a batch size of 32, resulting in a global batch size of 1024 generations per gradient step. Following (Liu et al., 2025), we use a mini-batch size of 256 and perform 4 gradient updates per rollout step.

**Operator-consistent RL with PDR.** For training, we use the **PDR** operator with configuration (4 parallel generations, 1 round) $g = [4]; k = 2$, and use the training objective described in Equation (11) and make two changes to the baseline RL method above: (i) increasing the input prompt length from 2048 tokens to 10240 to allow for the compact workspace to be part of the input, and (ii) mixing the standard RL and operator RL batches in the dataloader, keeping all other design choices the same. This setup allows to scale inference compute within the RL training.

**Results.** Table 2 summarizes the main results. The resulting model from each RL objective is evaluated for Long CoT generation and **PDR**. **PDR** RL improves over the baseline by $+3.34$ points on AIME 2024 and $+1.67$ points on AIME 2025. With continual updates starting from a baseline RL checkpoint, additional **PDR** RL yields larger gains of $+5.00$ and $+4.59$ percentage points on AIME 2024 and AIME 2025, respectively. Additionally, we also observe marginal gains on Long CoT generations with **PDR** RL training. These results indicate that training with operator-consistent RL objectives reduces the mismatch between training and deployment, converting extra compute into accuracy without increasing the per-call sequential budget.

## 4 CONCLUSION

The paper initiates exploration of a broader design space around "long CoT." We study in detail two operators, in this design space, **SR** and **PDR** which give better accuracy on benchmark tasks than well-known commercial models using standard long CoT, while offering the benefit of smaller context size. Empirically, compact-memory iteration outperforms long-trace baselines at matched $B_{\text{seq}}$. Shallow PDR yields the largest gains (e.g., +10% on AIME 2024 and +11% on AIME 2025), showing that evidence accumulation via bounded summaries can substitute for long reasoning traces while holding latency fixed. Beyond inference orchestration, aligning *training with inference* using an *operator-consistent RL* objective further improves performance (e.g., $\sim 5\%$ on AIME 2024 and AIME 2025), suggesting that models can learn the meta-skills (verify, refine, compress, diversify) that make iteration effective.

Promising future directions include learning to improve the synthesis operator $\mathcal{D}$ (trainable summaries), adaptive round and fan-out schedules conditioned on uncertainty, budget-aware controllers for allocating test-time compute, tighter integration with verifiers and tool use. We also see value in scaling studies and cross-domain evaluations (reasoning, coding, planning) to map when short-context iteration most benefits accuracy and latency.

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

## A    RELATED WORK

**Test-time reasoning with long traces.** Eliciting step-by-step "chains of thought" improves accuracy on multi-step tasks (Wei et al., 2022). Recent "reasoning" models (e.g., OpenAI *o1*) explicitly trade more test-time thinking for better results, increasing tokens and latency (Jaech et al., 2024). Sampling multiple traces and aggregating answers (*self-consistency*) further boosts performance but scales cost with the number of samples (Wang et al., 2023). Our approach targets the opposite design point: keep each call short while letting evidence accumulate across rounds via a bounded summary.

**Iterative self-improvement.** A growing line of work lets models critique and refine their own outputs: *Self-Refine* alternates self-feedback and revision (Madaan et al., 2023); *Reflexion* maintains a textual memory of reflections to guide subsequent attempts (Shinn et al., 2023); *CRITIC* verifies with tools and revises accordingly (Gou et al., 2023). Multi-agent debate improves factuality and reasoning via argumentation and adversarial checking (Irving et al., 2018; Du et al., 2023b). Our operator shares the self-improvement spirit but constrains per-call context by using a *round-wise, re-synthesized* workspace $C^{(r)}$ instead of replaying full histories.

**Structured search beyond single chains.** Prompting schemes structure exploration explicitly: *Tree of Thoughts* (ToT) (Yao et al., 2023) expands and evaluates branches of reasoning; *Graph of Thoughts* (Besta et al., 2024) generalizes to arbitrary thought graphs ; *Least-to-Most* (Zhou et al., 2022) decomposes problems into subproblems solved in sequence. These methods typically grow tokens with breadth/depth or rely on long contexts to carry intermediate state. In contrast, PDR concentrates exploration early within a round, then distills to a bounded $C^{(r)}$.

**Multi-agent debate and compressed debate.** Multi-agent debate improves factuality and reasoning by having multiple LLM "agents" propose answers and iteratively read and critique one another's responses, converging to a final answer (Du et al., 2023a). Our *Parallel–Distill–Refine (PDR)* can be viewed as a *compressed* debate: treat the round's diverse drafts as agent outputs, but instead of replaying full transcripts to every agent, we distill them into a bounded, round-wise workspace $C^{(r)}$ and condition the next generation on $C^{(r)}$. This preserves the benefits of cross-agent scrutiny while controlling per-call context and both $B_{seq}$ and $B_{total}$.

**Compact summaries vs. persistent memory.** Agent systems often add external memory or retrieval to persist context across sessions or tasks; thought buffers and memory-augmented agents exemplify this direction (e.g., Buffer-of-Thoughts) (Yang et al., 2024). We instead use *non-persistent*, round-wise summaries: $C^{(r)}$ is freshly synthesized from current drafts, capturing agreements, contradictions, intermediate results, and open subgoals, while retiring stale notes. This choice keeps prompts short and reduces long-context failure modes (Liu et al., 2023).

**Training signals for reasoning and alignment with inference.** Process-level supervision (*verify step-by-step*) can train models to produce reliable intermediate steps, and process reward models enable test-time search and verification (Lightman et al., 2023). Preference-based alignment methods (RLHF/DPO/RLAIF) are widely used but typically optimize for single-pass outputs rather than iterative read/write operators (Rafailov et al., 2023; Bai et al., 2022). We address this train–test mismatch by optimizing an *operator-consistent* objective: roll out the same read/write interface used at inference and reward effective updates to $C^{(r)}$ and the artifact.

**Budget-aware evaluation and test-time compute.** Recent work argues for comparing methods at matched compute budgets and reporting token usage, not just accuracy (Wang et al., 2024). Our protocol reports *sequential budget* $B_{seq}$ (latency proxy along the accepted path) and *total budget* $B_{total}$ (all tokens, including discarded branches), enabling apples-to-apples comparisons among single-pass, long-trace, sampling-heavy, and iterative pipelines.

**Global workspace and modular coordination.** Our use of a compact, round-wise summary $C^{(r)}$ is conceptually related to the *shared global workspace* proposed by Goyal et al. (2022), which enables coordination among neural modules through a small communication bottleneck (inspired by Global Workspace Theory (Baars, 2005; Shanahan, 2006)). In contrast, our workspace is *textual*, *re-synthesized each round* rather than persisted, and used as an *inference-time operator* for short-context iteration with explicit token budgets ($B_{seq}$, $B_{total}$). Thus, we borrow the coordination intuition while avoiding long-context replay and architectural changes.

## A.1 PROMPTS

### A.1.1 SEQUENTIAL REFINEMENT

> **Refinement Prompt**
>
> ```
> Solve the following math problem efficiently and clearly.
>     Please reason step by step, and put your final answer
>     within $\\boxed{answer}$.
>
> Where [answer] is just the final number or expression that
>     solves the problem.
>
> Problem: {{ problem }}
>
> Here is an example candidate response wrapped in angle
>     brackets:
>
> <model_response>
> # Solution response from previous turn
> </model_response>
>
> Treat the response as unverified; and refine the response
>     without starting from scratch to come up with the best
>     answer.
> ```

### A.1.2 PARALLEL-DISTILL-REFINE

> **Refinement Prompt (Non-summary)**
>
> ```
> Solve the following math problem efficiently and clearly.
>     Please reason step by step, and put your final answer
>     within $\\boxed{answer}$.
>
> Where [answer] is just the final number or expression that
>     solves the problem.
>
> Problem: {{ problem }}
>
> Here are some candidate responses, each wrapped in angle
>     brackets:
>
> <model_response_1>
> # Solution response from previous round
> </model_response_1>
>
>
> ...
>
> <model_response_k>
> # Solution response from previous round
> </model_response_k>
>
> Treat the responses from previous round as unverified; and
>     uses these responses without starting from scratch to come
>     up with the best answer.
> ```

**Refinement Prompt (Summary)**

```
Solve the following math problem efficiently and clearly.
    Please reason step by step, and put your final answer
    within $\\boxed{answer}$.

Where [answer] is just the final number or expression that
    solves the problem.

Problem: {{ problem }}

Here is a summary of the reasoning traces by a few other
    solvers:

<summary>
# Summary of all solutions from the previous iteration
</summary>

Treat the response as unverified; and refine the response
    without starting from scratch to come up with the best
    answer.
```

**Summary Prompt**

```
You are tasked with aggregating multiple reasoning traces into
    a single, cohesive explanation or summary.

Below are {{ width_id }} candidate responses, each wrapped in
    angle brackets and numbered in order of appearance:

{{ solutions_block }}

Your goal is to identify common themes, reconcile differences,
    and synthesize the information into a unified response.

Be sure to preserve key insights from each trace and ensure
    the final output is logically consistent and comprehensive.

Avoid discarding unique or contradictory insights; highlight
    and address them where possible.
```

## A.2 TRAINING SETUP

We run a very small SFT stage on the pre-trained 8B base model using a batch size of 2M tokens, max sequence length of 32768, and a learning rate of $2 \times 10^{-5}$ using the AdamW optimizer (Loshchilov & Hutter, 2017) on 32 H100 GPU nodes for approximately 4 epochs and 8B tokens in total. For RL, we use a constant learning rate of $5 \times 10^{-7}$, AdamW optimizer (Loshchilov & Hutter, 2017) with $\epsilon = 10^{-15}$, weight decay of 0.01, and a linear warmup of 100 steps. We use 80 Nvidia H200 GPUs for the baseline RL run with a 64/16 generators/trainers split and 288 H200 GPUs for **PDR** and continual **PDR** RL with a 256/16 generators/trainers split to parallelize inference during rollout generation. We run all RL training for 800 steps.

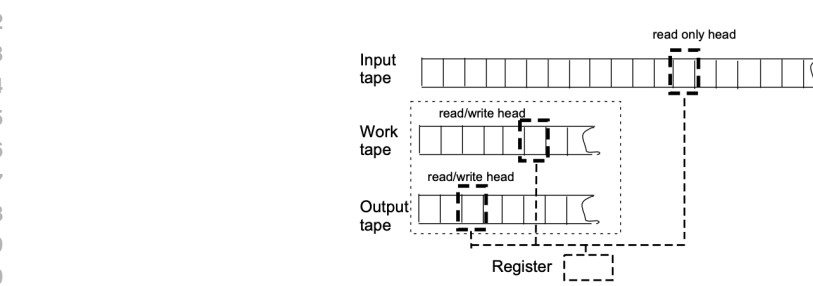

Figure 5: Space-bounded Turing Machine [Figure from *Computational Complexity* by Arora and Barak, 2007]. The input has size $N$ and the machine has read-only capability for the input. A special "tape head" can be moved over the input to read bits from it. The amount of working memory (read/write/erase) for actual computation has size $S(N)$ where $S(N) \geq \log N$.

## B    COMPLEXITY OF SPACE-BOUNDED COMPUTATION

Our work focused on language models that emit reasoning traces that are longer than the context size. We sketch similarity to the setting *space-bounded computation* which is formally studied in computational complexity theory.

Since LLMs have probabilistic output (unless if temperature is set to 0) the fixed-context LLM considered in the paper is most similar to randomized space-bounded machine.

The most interesting result about randomized space-bounded machines is that if the input contains a graph of $N$ vertices, then the randomized machine can determine connectivity of the $N$-vertex graph even though it only has $O(\log N)$ space.

Furthermore, imagine that the graph of size $N$ is a knowledge-graph whose local structure is known to the space-bounded machine. Specifically, given vertex indices $i, j$ the space-bounded machine is able to determine whether edge $\{i, j\}$ exists in the graph. Then the machine does not need access to the full graph tape at all! It can do a random walk through the graph "in its mind" to determine connectivity. This is the setting closest to ours, whereby seemingly complex reasoning– connectivity of an $N$-node graph–can be carried out in much less than $N$ space.

## C    ADDITIONAL RESULTS

### C.1    ORACLE

Similar to Figure 4, we observe that having incorrect solutions in the context workspace can heavily degrade performance, and this effect is more noticeable for `o3-mini`.

### C.2    **SR** AND **PDR** OPERATORS

AIME 2025 results using the two iterative improvement operators **SR** and **PDR** are presented in Figure 7. For sequential token budget of $49k$, the performance on `o3-mini` improves from 73.5 for Long CoT to 77.1 using **SR** operator and further to 82.9 using the **PDR** operator.

Additionally, we show two **SR** variant results in Table 3, where error analysis followed by solution generation leads to improvements on `o3-mini` without any affect on the sequential token budget $B_{\text{seq}}$.

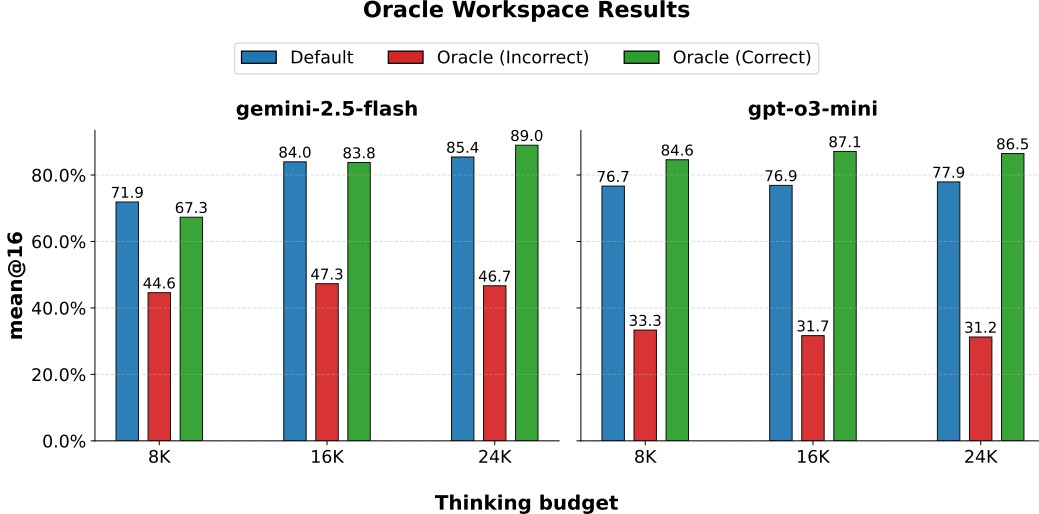

Figure 6: **AIME 2025: Anchoring bias due to +ve and −ve examples:** With **PDR** we compare three selection policies for the summary: Random-$k$, Oracle-Incorrect (all $k$ candidates are incorrect), and Oracle-Correct (all $k$ candidates are correct), evaluated on both `gemini-2.5-flash` and `o3-mini`. Across all thinking budgets, admitting only incorrect candidates into the summary yields a pronounced drop in accuracy, whereas admitting only correct candidates improves over the Random-$k$ baseline. The degradation under Oracle-Incorrect is markedly larger for `o3-mini` than for `gemini-2.5-flash`, indicating weaker self-verification in `o3-mini`.

Table 3: **SR operator variants**: Instead of just asking the model to refine the solution, we also ask the model to find and analyze errors in the solution followed by the correct solution. Error analysis followed by solution generation leads to better performance for `o3-mini` but not for `gemini-2.5-flash`.

| Model | Benchmark | Thinking Budget | SR | SR-*Error* |
|---|---|---:|---|---|
| `gemini-2.5-flash` | AIME 2024 | 24576 | 88.75 | 87.71 |
| `gemini-2.5-flash` | AIME 2025 | 24576 | 78.75 | 79.17 |
| `gpt-o3-mini` | AIME 2024 | 24576 | 80.83 | 82.08 |
| `gpt-o3-mini` | AIME 2025 | 24576 | 73.13 | 77.92 |

## C.3 TOKEN MATCHED BASELINES

In Figure 8, we run the **SR** operator for very high depths to check at what $B_{\text{seq}}$ budget, it matches the performance of the **PDR** operator. **SR** needs a budget of $442k$ tokens to match the performance of **PDR** on $172k$ tokens.

## D REPRODUCIBILITY STATEMENT

While we do not release our source code, we provide all information necessary to independently reproduce our results. For example, the inference setup for both API models `gemini-2.5-flash` and `o3-mini` is detailed in Section 3. The prompts for **SR** and **PDR** operators are detailed in Appendix A.1.1 and Appendix A.1.2. All the operator configurations are detailed in Section 3.

For SFT and RL training, the dataset and training details along with hyper-parameters are described in Section 3 and Appendix A.2, respectively. Together, these references specify our assumptions and experimental details with sufficient precision for a successful reproduction of our results.

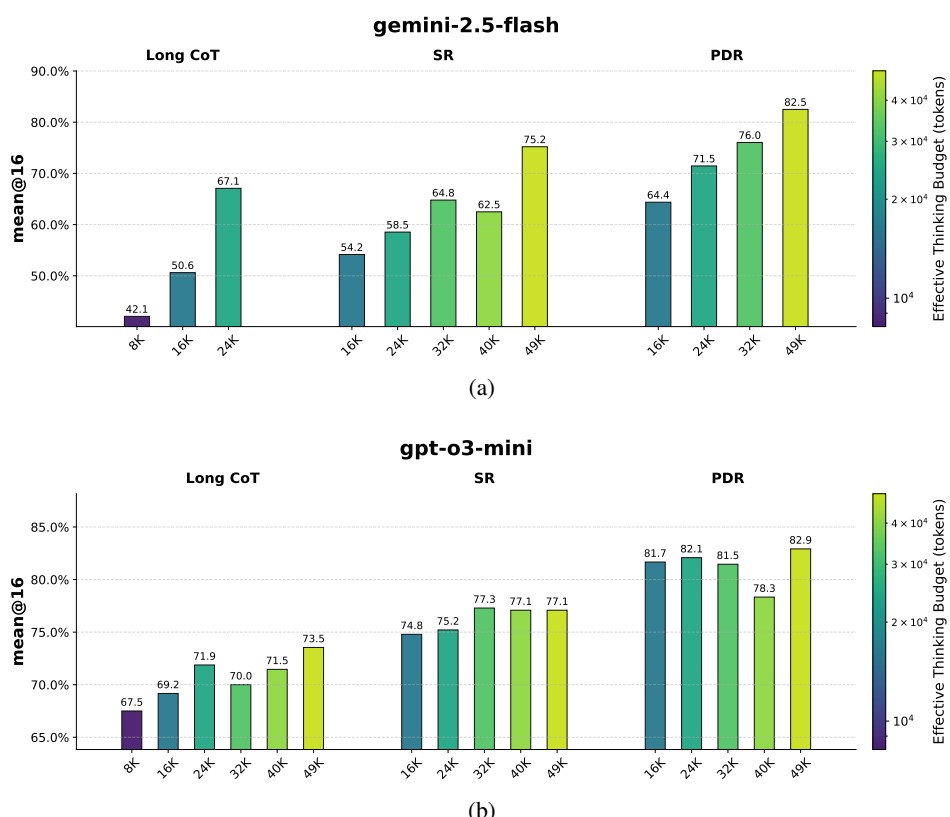

(a)

(b)

Figure 7: **AIME 2025: Iterative improvement beats single-pass long-CoT at matched sequential budgets.** The $x$-axis reports $B_{\mathrm{seq}}$: the thinking tokens consumed along the accepted path of the iterative chain, plus any distilled summary that conditions the next step. Tokens spent on unused parallel proposals are excluded, so $B_{\mathrm{seq}}$ serves as a latency proxy. At comparable $B_{\mathrm{seq}}$, both **SR** and **PDR** outperform the single-pass long CoT baseline, with **PDR** yielding the largest gains by converting additional total compute (via parallelism) into accuracy without increasing per-call context.

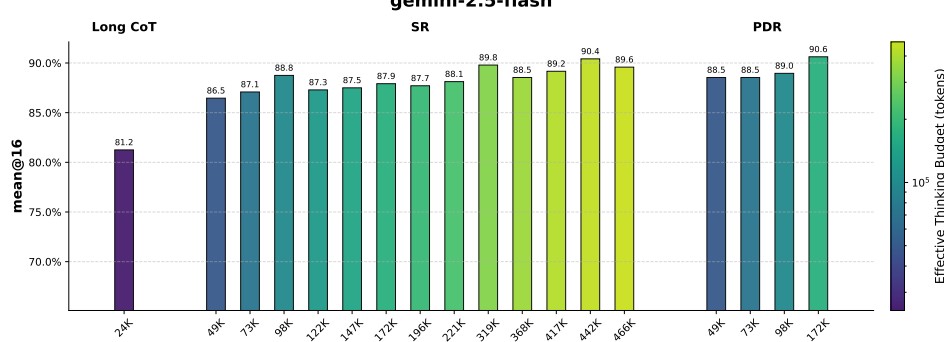

Figure 8: **AIME 2024: Iterative improvement beats single-pass long-CoT at matched sequential budgets.** The $x$-axis reports $B_{\mathrm{seq}}$: the thinking tokens consumed along the accepted path of the iterative chain, plus any distilled summary that conditions the next step. Tokens spent on unused parallel proposals are excluded, so $B_{\mathrm{seq}}$ serves as a latency proxy. At comparable $B_{\mathrm{seq}}$, both **SR** and **PDR** outperform the single-pass long CoT baseline, with **PDR** yielding the largest gains by converting additional total compute (via parallelism) into accuracy without increasing per-call context.

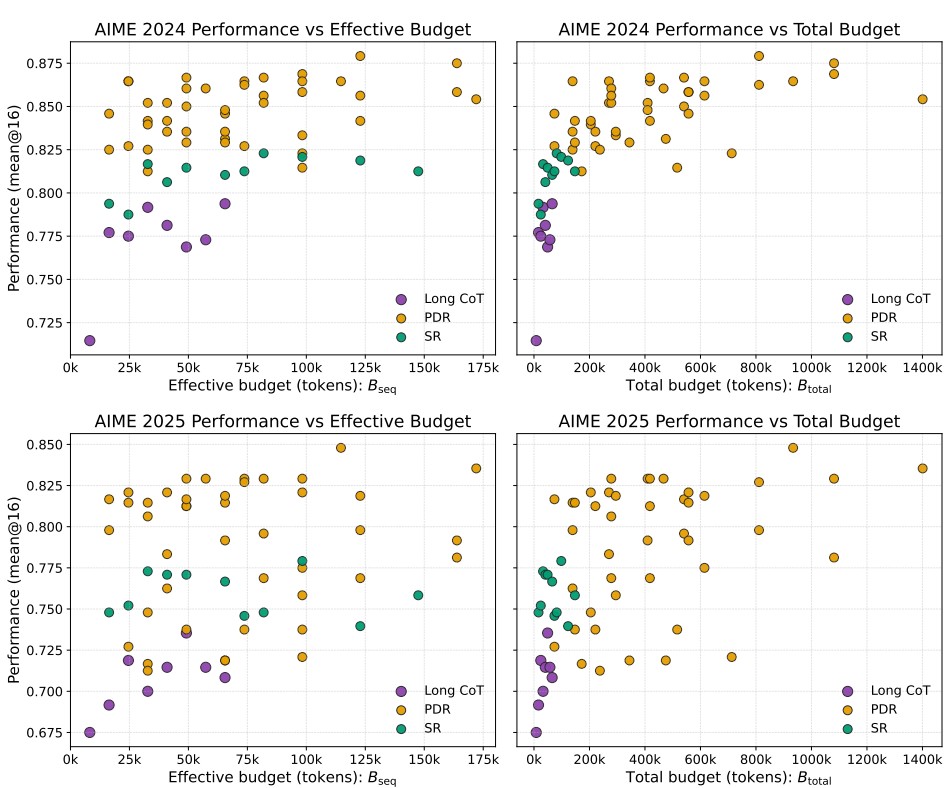

Figure 9: **Token Budgets comparison:** We plot all the different configurations for Long CoT, **SR** and **PDR** operators for both $B_{seq}$ and $B_{total}$ token budgets for o3-mini. For both $B_{seq}$ and $B_{total}$, **PDR** forms the pareto-frontier and gives consistent gains over Long CoT and **SR**.

