# OpenReview forum: "Large Language Models as Improvement Operators: Better Reasoning by Iteration"
_ICLR.cc/2026/Conference — Submitted to ICLR 2026_

### Official Review · Reviewer_7aGP · 2025-10-30

**Soundness:** 3
**Presentation:** 1
**Contribution:** 3
**Rating:** 4
**Confidence:** 3

**Summary:**

This paper reframes reasoning with large language models (LLMs) as an iterative improvement process, introducing the concept of LLMs as improvement operators.

Two inference operators are proposed:
**Sequential Refinement (SR)** — iteratively refines a single answer in short contexts.
**Parallel–Distill–Refine (PDR)** — generates multiple drafts, summarizes them into a bounded workspace, and refines based on this summary iteratively.

To reduce the train–test mismatch between long chain-of-thought (CoT) training and short-context iterative inference, the authors propose operator-consistent reinforcement learning (Op-RL), aligning training and inference through rollouts that mimic the generate–distill–refine interface.

Experiments on AIME 2024 and 2025 math benchmarks show that PDR achieves up to +10–11% accuracy gains over long CoT under comparable latency budgets. The operator-consistent RL further provides a ~5% boost.

**Strengths:**

**Conceptually novel framing** Viewing LLMs as “improvement operators” provides a fresh and systematic perspective for reasoning optimization.

**Inference efficiency focus** The work directly tackles the trade-off between reasoning accuracy and token cost, a central issue in test-time scaling.

**Operator-consistent RL** A meaningful step toward aligning training and inference under short-context reasoning.

**Clear experimental protocol** The paper consistently reports both sequential budget (B_seq) and total budget (B_total), encouraging cost-aware evaluation.

**Weaknesses:**

**Limited evaluation scope (Major Concern)**
The empirical validation relies solely on AIME 2024/2025, two relatively narrow math benchmarks.
There is no evidence of generalization to other reasoning domains (e.g., commonsense, logic puzzles, code generation, multimodal reasoning).
Given the paper’s broad claim of “better reasoning by iteration,” this narrow scope weakens the empirical foundation.

**Cost analysis is incomplete**
Although the paper introduces B_seq (latency proxy) and B_total (compute proxy), there is little quantitative discussion on real compute cost (e.g., wall-clock time, GPU-hours, or energy).
The operator-consistent RL training uses large-scale setups (80–288 H200 GPUs), but the paper never clarifies whether the performance gains justify this cost.

**Reproducibility concerns**
Experiments rely on closed-source models (Gemini 2.5 Flash, GPT-o3-mini), and the open 8B model results are limited to internal evaluation.

**Questions:**

1. Can the proposed PDR method generalize beyond math tasks (e.g., planning, commonsense reasoning, code)?

2. What is the real inference-time speedup compared to long CoT, when measured in latency and wall-clock compute (not just token counts)?

---

> ### Author Response · Authors · 2025-11-25
> **Response to Reviewer 7aGP**
>
> We thank the reviewer for their feedback and we address the weaknesses below. We also report additional experimental results.
>
> > Limited evaluation scope (Major Concern) The empirical validation relies solely on AIME 2024/2025, two relatively narrow math benchmarks. There is no evidence of generalization to other reasoning domains (e.g., commonsense, logic puzzles, code generation, multimodal reasoning). Given the paper’s broad claim of “better reasoning by iteration,” this narrow scope weakens the empirical foundation.
>
> **Response**: We agree that evaluating only on the AIME benchmark is too narrow. To address this, we now include two additional, substantially larger, non-math benchmarks in the paper:
>
> - GPQA (diamond), which contains graduate-level questions in physics, biology, and chemistry.
> - LiveCodeBench (2025 split), which consists of competitive programming problems from platforms such as LeetCode, AtCoder, and Codeforces.
>
> In addition to the commercial API models, we also report results for the open-weight GPT-OSS models on these benchmarks to support reproducibility and to demonstrate that the benefits of PDR are not tied to a single math dataset or to proprietary APIs.
>
> | | LiveCodeBench | LiveCodeBench | GPQA (diamond) | GPQA (diamond) |
> | :--- | :---: | :---: | :---: | :---: |
> | | Long CoT | PDR ($w=[8], k=4$) | Long CoT | PDR ($w=[8], k=4$) |
> | gemini-2.5-flash | 55.96 | **61.25** | 75.98 | **82.32** |
> | gpt-o3-mini | 48.90 | **58.61** | 73.30 | **76.26** |
> | GPT-OSS-20B | 55.62 | **63.99** | 61.55 | **68.50** |
> | GPT-OSS-120B | 63.12 | **69.71** | 69.92 | **72.29** |
>
> Across both benchmarks, PDR consistently improves over the long CoT baseline for all models, suggesting that its benefits extend beyond math to other domains as well.
>
> > Reproducibility concerns Experiments rely on closed-source models (Gemini 2.5 Flash, GPT-o3-mini), and the open 8B model results are limited to internal evaluation.
>
> **Response**: We added results for the GPT-OSS model family in the table above to support reproducibility. We also include additional experiments with the open-weight Qwen3 family of models in the table below.
>
> | | AIME 2025 | AIME 2025 |
> | :--- | :---: | :---: |
> | | Long CoT | PDR ($w=[8, 8], k=4$)
> | Qwen3-4B-Instruct-2507 | 43.13 | 56.67 |
> | Qwen3-30B-A3B-Instruct-2507 | 51.04 | 69.17 |
>
> We are currently running further Qwen3 experiments at additional model sizes and will update the paper with the complete size ablation.
>
> The additional results highlight the benefits of our proposed approaches beyond closed models and math benchmarks.
>
> > Cost analysis is incomplete Although the paper introduces B_seq (latency proxy) and B_total (compute proxy), there is little quantitative discussion on real compute cost (e.g., wall-clock time, GPU-hours, or energy).
>
> **Response**: We agree that our current presentation focuses on token-level budgets and does not spell out cost in units such as GPU-hours or wall-clock time. Our intent with $B_{seq}$ and $B_{total}$ is to provide hardware-agnostic proxies:
>
> - $B_{seq}$ controls the *per-call context length* and therefore strongly correlates with per-query latency on a fixed system.
> - $B_{total}$ is proportional to the total number of tokens generated/read, and thus to overall compute / GPU-hours.
>
> For a fixed model and inference stack, wall-clock latency is approximately linear in $B_{seq}$, and total energy / GPU cost is approximately linear in $B_{total}$. Since all decoding strategies (SR, long CoT, PDR) are implemented with the same backend, comparing them at matched $B_{seq}$ and $B_{total}$ corresponds to comparing them at roughly matched latency and compute cost. We will make this connection explicit in the paper and add a short discussion of how to translate from token budgets to approximate wall-clock latency on typical hardware.
>
> > The operator-consistent RL training uses large-scale setups (80-288 H200 GPUs), but the paper never clarifies whether the performance gains justify this cost.
>
> **Response**: Both the baseline RL and our operator-consistent RL runs use the same large-scale setup (80-288 H200 GPUs). Our goal in these experiments is not to claim that such scale is necessary to use PDR, but to show that PDR remains effective and stable in a realistic, large-scale training regime. In smaller-scale preliminary runs, we find that reducing the number of trainers/generators and compensating with more RL steps preserves most of the gains, at the cost of increased wall-clock training time. We will clarify this trade-off in the revised version of the paper.
>
> Finally, we emphasize that the PDR inference procedure itself is model-agnostic and can be applied to off-the-shelf models (including commercial APIs and open-weight models) with no additional training cost. The large-scale RL experiments should be viewed as a "best case" showing that training the model to be operator-consistent further improves PDR, not as a prerequisite for benefiting from PDR.

---

> > ### Comment · Reviewer_7aGP · 2025-11-26
> > **Response to the authors**
> >
> > The rebuttal solved most of my concerns. However, I am still concerned about the computational cost and time cost PDR compared with Long CoT.
> >
> > Also, I suggest that the authors open-source all the code, add the rebuttal experiments in their final version.

---

> > > ### Author Response · Authors · 2025-11-27
> > > **Followup response to reviewer 7aGP**
> > >
> > > We thank the reviewer for responding to our rebuttal. We address the remaining questions below:
> > >
> > > > However, I am still concerned about the computational cost and time cost PDR compared with Long CoT.
> > >
> > > **Response**: We do a simple inference profiling test using an 8B model on 1 H100 GPU for Long CoT and PDR. We use an input prompt length of 8192 tokens, batch size of 1 for Long CoT. We use an output generation length ranging from 1k-73k tokens and report the latencies below. For PDR, we use batched computation and set batch size of 8 for generation, and output generation length of 8k tokens. We do 5 measurements for each setting, and report the average latencies.
> > >
> > > | Generation Length | Batch size | TTFT (seconds) | Total time (seconds) |
> > > | :---: | :---: | :---: | :---: |
> > > | 1024 | 1 | 0.1637 | 12.6808 |
> > > | 2048 | 1 | 0.1640 | 25.2173 |
> > > | 8192 | 1 | 0.1647 | 101.3881 |
> > > | 16384 | 1 | 0.1714 | 202.7761 |
> > > | 32768 | 1 | 0.1795 | 412.5801 |
> > > | 65536 | 1 | 0.1976 | 830.8500 |
> > > | 73728 | 1 | 0.2012 | 938.6954 |
> > > | 8192 | 8 | 1.3354 | 113.8501 |
> > >
> > > As we can see from the above table, a batched computation of 8 parallel branches (last row) takes 113.8501 - 101.3881 = 12.462 seconds more (for generation budget of 8192 tokens), but it’s not significant given it’s using 8x more FLOPs and generating 8x more total tokens.
> > >
> > > We get the following latencies for Long CoT and PDR:
> > >
> > > | Configuration | Generation Budget | #LLM Calls | $B_{seq}$ | $B_{total}$ | Latency (s) |
> > > | :---: | :---: | :---: | :---: | :---: | :---: |
> > > | Long CoT | 16384 | 1 | 16384 | 16384 | 202.7761 |
> > > | Long CoT | 73728 | 1 | 73728 | 73728 | 938.6954 |
> > > | PDR ($w=[8], k=4$) | 8192 | 2 | 16384 | 73728 | 215.2382 |
> > >
> > >
> > > For PDR, the first LLM call takes 113.8501 seconds and the second call takes 101.3881 seconds (because batch size=1 during refinement in PDR), so a total of 215.2382 seconds. PDR (row 3) is processing more total tokens compared to Long CoT (row 1), but we are able to leverage GPU parallelization to keep similar latencies and much higher accuracy for PDR compared to Long CoT for a fixed $B_{seq}$. If we consider $B_{total}$, PDR (row 3) latency is much lower and we still get better performance in PDR because the Long CoT (row 2) performance stagnates after a certain generation budget.
> > >
> > > There's a tradeoff between compute spent (FLOPs) and accuracy, and PDR is able to extract out more performance by spending a lot more compute. The affect on latency is minor because of GPU parallelization.
> > >
> > > Additionally, this latency effect can be reduced further by deploying multiple model copies and reducing the batch size, which will decrease the minor latency gap between Long CoT and PDR while retaining the high performance of PDR compared to Long CoT.
> > >
> > > > Also, I suggest that the authors open-source all the code, add the rebuttal experiments in their final version.
> > >
> > > **Response**: We will include all the additional results from the rebuttal in the final version of the paper, and we are also writing a PDR/SR orchestration around a popular open-source eval harness used by the community (https://github.com/EleutherAI/lm-evaluation-harness) and will raise a PR to include the PDR/SR inference-time setup for evaluation.
> > >
> > >
> > > We are happy to answer any additional questions that the reviewer may have and welcome any suggestions to improve this work further.

---

> > > > ### Comment · Reviewer_7aGP · 2025-11-28
> > > > **Response to the authors**
> > > >
> > > > Thanks for the author's rebuttal. I will raise my score.

---

> > > > > ### Author Response · Authors · 2025-11-28
> > > > > **Thank you**
> > > > >
> > > > > We thank the reviewer for carefully considering our rebuttal and for the constructive feedback on how to further improve our work. We also appreciate your decision to raise the score. If there are any remaining concerns or additional suggestions, we would be very happy to address them.

---

### Official Review · Reviewer_RHtA · 2025-11-01

**Soundness:** 3
**Presentation:** 3
**Contribution:** 2
**Rating:** 4
**Confidence:** 3

**Summary:**

This paper proposes viewing LLMs as "improvement operators" that iteratively refine solutions using bounded context rather than single long chains of thought. The authors introduce two inference methods: Sequential Refinement (SR) - iteratively improving a single solution, and Parallel-Distill-Refine (PDR) - generating multiple parallel solutions, distilling them into a compact summary, then refining. They also propose "operator-consistent" training that aligns the training objective with this iterative inference pattern. Experiments on AIME 2024/2025 math problems show PDR can achieve better accuracy than long CoT baselines while maintaining lower latency and context size.

**Strengths:**

1. Viewing LLMs as improvement operators with explicit budget constraints (sequential vs. total tokens) is a useful perspective for understanding the accuracy/latency/cost tradeoffs. The methods are straightforward to implement using existing models via prompting, without architectural changes
2. The paper evaluates multiple distillation strategies, studies oracle experiments to understand verification capabilities, and measures both sequential and total token budgets.
3. PDR achieves +10-11% improvements on AIME benchmarks over long CoT at matched sequential budgets

**Weaknesses:**

1. The paper doesn't explain when short-context iteration helps vs. hurts. What problem properties make PDR effective? When should practitioners use this? No results on reasoning tasks without automated verification, coding, planning, or other domains
2. While per-call context is bounded, the *total* information processed grows with rounds. The paper conflates "bounded workspace per round" with "compact memory" but the system still processes increasing amounts of text
3. Only compares to single long CoT. Missing comparisons to: self-consistency with multiple samples, beam search, other structured reasoning methods (ToT, GoT). The operator RL uses only 1 round during training but multiple rounds at test time. Why this mismatch? How sensitive are results to this choice?

**Questions:**

**How does this compare to self-consistency or best-of-N sampling?** At matched total compute, is PDR actually better than just sampling multiple solutions and picking the best?

**What about the train-test mismatch in number of rounds?** Training uses 1 round, testing uses multiple. Did you experiment with multi-round training rollouts?

**Can you provide failure analysis?** When does PDR make things worse? What types of problems benefit most?

---

> ### Author Response · Authors · 2025-11-25
> **Response to Reviewer RHtA**
>
> We thank the reviewer for their feedback and address their concerns/questions below along with additional experiments:
>
> > No results on reasoning tasks without automated verification, coding, planning, or other domains.
>
> **Response**: We agree that evaluating only on the verifiable AIME benchmark is too narrow. To address this, we now include two additional, substantially larger, non-math benchmarks:
>
> - GPQA (diamond), which contains graduate-level questions in physics, biology, and chemistry.
> - LiveCodeBench (2025 split), which consists of competitive programming problems from platforms such as LeetCode, AtCoder, and Codeforces.
>
> In addition to the commercial API models, we also report results for the open-weight GPT-OSS models on these benchmarks to support reproducibility and to demonstrate that the benefits of PDR are not tied to a single math dataset or to proprietary APIs.
>
> | | LiveCodeBench | LiveCodeBench | GPQA (diamond) | GPQA (diamond) |
> | :--- | :---: | :---: | :---: | :---: |
> | | Long CoT | PDR ($w=[8], k=4$) | Long CoT | PDR ($w=[8], k=4$) |
> | gemini-2.5-flash | 55.96 | **61.25** | 75.98 | **82.32** |
> | gpt-o3-mini | 48.90 | **58.61** | 73.30 | **76.26** |
> | GPT-OSS-20B | 55.62 | **63.99** | 61.55 | **68.50** |
> | GPT-OSS-120B | 63.12 | **69.71** | 69.92 | **72.29** |
>
> Across both benchmarks, PDR consistently improves over the long CoT baseline, suggesting that its benefits extend beyond math to other domains as well.
>
> > While per-call context is bounded, the total information processed grows with rounds. The paper conflates "bounded workspace per round" with "compact memory" but the system still processes increasing amounts of text
>
> **Response**: We distinguish two different quantities: (i) the workspace / per-call context size, i.e., how many tokens the model conditions on at once (in the input prompt), and (ii) the total tokens processed across all rounds. Our use of “bounded workspace” and “compact memory” refers to (i), not to the cumulative compute in (ii).
>
> Concretely, in PDR we enforce a fixed memory workspace size across rounds. For example, if the workspace budget is 32,768 tokens, then at each round the model always sees the following input: (*problem* + *instructions*) (N tokens) + *workspace* ($\leq 32768$ tokens), so the per-call context is bounded by 32,768+N tokens (with N $\approx$ 512). Earlier drafts are not appended indefinitely; they must be summarized or filtered into this fixed workspace.
>
> > How does this compare to self-consistency or best-of-N sampling? At matched total compute, is PDR actually better than just sampling multiple solutions and picking the best?
>
> We add self-consistency results in the table below for both gemini-2.5-flash and gpt-o3-mini, and find PDR outperforms the self-consistency baseline as well.
>
> | | AIME 2024 | AIME 2024 | AIME 2024 | GPQA (diamond) | GPQA (diamond) | GPQA (diamond) |
> | :--- | :---: | :---: | :---: | :---: | :---: | :---: |
> | | Long CoT | $maj@16$ (Self Consistency) | PDR ($w=[8], k=4$) | Long CoT | $maj@16$ (Self-consistency) | PDR ($w=[8], k=4$) |
> | gemini-2.5-flash | 81.2 | 86.67 | 88.5 | 75.98 | 79.80 | 82.32 |
> | gpt-o3-mini | 76.25 | 83.33 | 86.7 | 73.30 | 74.24 | 76.26 |
>
> > The paper doesn't explain when short-context iteration helps vs. hurts. What problem properties make PDR effective? When should practitioners use this?
>
> **Response**: We did a manual inspection of some samples where PDR underperforms or matches standard CoT and found two interesting cases:
>
> - Few correct drafts among the N parallel samples: In these instances, only a small fraction of drafts are correct. Both top-$k$ and global summary then mix a large number of incorrect solutions with a tiny number of correct ones, without explicitly isolating the failure modes of the incorrect generations. Therefore, the summary does not reliably surface signals correlated with correctness. In such rare cases, a long and unconstrained CoT trajectory can sometimes follow the single correct reasoning path and outperform PDR, which is effectively averaging over mostly wrong trajectories.
>
> - No correct drafts among the N parallel samples: When all drafts are wrong, both long CoT and PDR predict incorrect answers. In principle, PDR should still be able to extract useful structure (partial progress, contradictions, etc.) but instead it over-weights a wrong pattern in subsequent rounds thus leading to summary drift (subsequent rounds refine and reinforce an incorrect line of reasoning) and wrong generations.
>
> More broadly, PDR’s gains are contingent on the model’s verification and refinement abilities: when these are weak, the distillation step can amplify incorrect patterns instead of correcting them, so PDR may not improve over standard CoT.

---

> > ### Author Response · Authors · 2025-11-25
> > **Response to Reviewer RHtA (Part 2)**
> >
> > > What about the train-test mismatch in number of rounds? Training uses 1 round, testing uses multiple. Did you experiment with multi-round training rollouts?
> >
> >
> > **Response**: It is correct that, for stability and cost reasons, we only use a one-round PDR rollout during RL. Conceptually, the policy we train is the per-round improvement operator: given a prompt and the current memory workspace, generate a refined attempt. Multi-round PDR at inference is just repeated application of this learned local operator. The model gets better at summarization and refinement due to the PDR training objective (Equation 11) even during this single round training. Due to this, we observe that single-round PDR RL improves both the single-round and multi-round evaluation scores.
> >
> > Unrolling several PDR rounds inside RL would multiply rollout cost by the number of rounds, which can improve the multi-round evaluations further due to more compute spent in RL.

---

### Official Review · Reviewer_gfiH · 2025-11-02

**Soundness:** 3
**Presentation:** 3
**Contribution:** 3
**Rating:** 6
**Confidence:** 4

**Summary:**

The authors argue that using the standard approach of eliciting long CoT represents just one point on the Pareto frontier (there are also other factors along with answer accuracy – context length, compute cost and latency) and explore two inference methods that frame LLM as an “improvement operator” – Parallel-Distill-Refine (PDR) and Sequential Refinement (SR). They also introduce RL training to address the potential train and inference-time mismatch with this new inference method. The evaluation which is done on AIME 2024 & AIME 2025 (which are math reasoning benchmarks), reveals that PDR in particular surpasses performance on long CoT baselines (for a given latency). Shallow PDR achieved gains of +10% on AIME 2024 and +11% on AIME 2025. RL training further improved these results.

**Strengths:**

1. The core idea of moving beyond a single reasoning trace towards other options on the Pareto Frontier makes complete sense.

2. PDR method is a good alternative for the problem of iterative refinement and using bounded, round-wise summary is innovative.

3. Budget evaluation which was done by making a distinction between sequential budget (latency proxy) and total budget (compute/cost proxy) is appreciated.

4. Impressive Results on AIME.

**Weaknesses:**

1. While the title mentions suggests reasoning in general, the evaluations are only on math tasks with verifiable rewards. Applicability to other settings remains unknown.

2. The authors haven’t provided details of AIME datasets in the paper, but to the best of my knowledge they contain around 30 questions each (?) – if so, then 10% gain would mean improvement on 3 out of 30 questions. It would have been better if the authors evaluated the approach on more datasets or datasets of larger size.

3. During RL training, the authors use a one-round PDR rollout. However, at inference, multi-round is done. Does this not create its own train-test mismatch?

**Questions:**

See Weaknesses

---

> ### Author Response · Authors · 2025-11-25
> **Response to Reviewer gfiH**
>
> We thank the reviewer for their positive outlook on our paper and highlighting the strengths of this work. We address the weaknesses/questions raised by the reviewer below:
>
> > the evaluations are only on math tasks with verifiable rewards. Applicability to other settings remains unknown. It would have been better if the authors evaluated the approach on more datasets or datasets of larger size.
>
> **Response**: We agree that evaluating only on the AIME benchmark is too narrow. To address this, we now include two additional, substantially larger, non-math benchmarks in the paper:
>
> - GPQA (diamond), which contains graduate-level questions in physics, biology, and chemistry.
> - LiveCodeBench (2025 split), which consists of competitive programming problems from platforms such as LeetCode, AtCoder, and Codeforces.
>
> In addition to the commercial API models, we also report results for the open-weight GPT-OSS models on these benchmarks to support reproducibility and to demonstrate that the benefits of PDR are not tied to a single math dataset or to proprietary APIs.
>
> | | LiveCodeBench | LiveCodeBench | GPQA (diamond) | GPQA (diamond) |
> | :--- | :---: | :---: | :---: | :---: |
> | | Long CoT | PDR ($w=[8], k=4$) | Long CoT | PDR ($w=[8], k=4$) |
> | gemini-2.5-flash | 55.96 | **61.25** | 75.98 | **82.32** |
> | gpt-o3-mini | 48.90 | **58.61** | 73.30 | **76.26** |
> | GPT-OSS-20B | 55.62 | **63.99** | 61.55 | **68.50** |
> | GPT-OSS-120B | 63.12 | **69.71** | 69.92 | **72.29** |
>
> Across both benchmarks, PDR consistently improves over the long CoT baseline, suggesting that its benefits extend beyond math to other domains as well.
>
> > The authors haven’t provided details of AIME datasets in the paper, but to the best of my knowledge they contain around 30 questions each (?) – if so, then 10% gain would mean improvement on 3 out of 30 questions
>
> **Response**: Yes, AIME has 30 problems per year (60 in total for 2024/2025). We report all evaluation results by averaging over 16 generations per sample, meaning that the simulated benchmark size is effectively 480 (30 $\times$ 16). But, we agree with the reviewer that AIME is quite small, and therefore, we report performance on much larger benchmarks in the table above (LiveCodeBench and GPQA).
>
> > During RL training, the authors use a one-round PDR rollout. However, at inference, multi-round is done. Does this not create its own train-test mismatch?
>
> **Response**: It is correct that, for stability and cost reasons, we only use a one-round PDR rollout during RL. Conceptually, the policy we train is the per-round improvement operator: given a prompt and the current memory workspace, generate a refined attempt. Multi-round PDR at inference is just repeated application of this learned local operator. The model gets better at summarization and refinement due to the PDR training objective (Equation 11) even during this single round training. Due to this, we observe that single-round PDR RL improves both the single-round and multi-round evaluation scores.
>
> We hypothesize that multi-round PDR training can improve the multi-round evaluations further due to more compute spent in RL.

---

### Official Review · Reviewer_79KF · 2025-11-03

**Soundness:** 2
**Presentation:** 3
**Contribution:** 2
**Rating:** 4
**Confidence:** 4

**Summary:**

This paper proposes a new way to think about large language model (LLM) inference. Instead of generating a single long chain of thoughts, it treats reasoning as an iterative improvement process that works in short rounds. The model repeatedly improves or refines earlier answers within a limited token budget.

Two main operators are introduced.
1. Sequential Refinement (SR): the model refines one draft step by step in short rounds, similar to incremental self-reflection.
2. Parallel Distill Refine (PDR): in each round, the model produces several candidate drafts, then summarizes or selects the most useful parts within a fixed workspace limit (κ tokens). This summary acts as shared context for the next refinement round.

The paper also introduces operator-consistent reinforcement learning (RL), which trains the model using the same short iterative process used during inference. This alignment helps the model learn how to improve solutions step by step instead of relying only on long reasoning traces. Experiments on math reasoning benchmarks such as AIME 2024 and AIME 2025 show that PDR and SR outperform long chain-of-thought (CoT) reasoning when measured under equal latency or token budgets. PDR achieves higher accuracy while using shorter reasoning paths. The paper also compares different workspace strategies, such as global summaries and top-k selections, and finds that summarizing or choosing top responses helps more than random selection.

**Strengths:**

1. The idea of treating inference as an iterative improvement process is simple and practical.
2. The paper provides a clear framework and connects reasoning to space-bounded or compressed computation and experiments are well controlled using clear token budgets, making comparisons fair.
3. The operator-consistent RL improves reasoning performance and aligns training with inference.

**Weaknesses:**

1. Evaluation is restricted to math problems only (AIME 2024, 2025) where rule-based verifiers (sympy, math-verify) exist. The generalizability of these findings to non-math domains remains unclear. Math problems may have unique properties (e.g., objective ground truth, verification tractability) that make PDR effective in ways that don't transfer.

2. The distillation operator D is critical as it compresses multi-draft solutions into a bounded workspace. Further, the paper explores four heuristic distillation strategies (global summary, top-k, random-k). However, it does not provide principled criteria for when each strategy should apply, and explain why global summary and per-sample top-k succeed whereas random-k underperforms.

3. Evaluation on commercial APIs (gemini-2.5-flash, o3-mini) limits reproducibility and ablations. The authors cannot, for example, vary model size systematically or inspect model internals. Further, an ablation across model sizes is not shown. Typically, 8B dense model trained with operator RL is relatively small by modern standards. There should have been an evaluation on larger models, say 20B+ to show effectiveness of the approach.

**Questions:**

1. Did you try an ablation on the mixing ratio? Try 25-75, 40-60, 60-40 splits in Equation 11. Does the 50-50 choice matter?

2. Can you describe scenarios where PDR underperforms: (1) Very short thinking budgets (do summaries compress too aggressively)? (2) Tasks requiring deep state tracking (does round-wise forgetting hurt)?

3. Figure 3 and Figure 9 show PDR dominates Long CoT on B_seq but SR dominates on B_total. This is stated but not deeply analyzed: (1) When would practitioners prefer SR (high latency tolerance, cost-constrained) vs. PDR? (2) How do parallelism constraints on typical hardware affect the tradeoff?

4. How does performance change with different workspace sizes?

---

> ### Author Response · Authors · 2025-11-25
> **Response to Reviewer 79KF**
>
> We thank the reviewer for their detailed review and highlighting the strengths of our submission. We provide the response to the weaknesses/questions along with additional experiments below:
>
> > Evaluation is restricted to math problems only (AIME 2024, 2025) where rule-based verifiers (sympy, math-verify) exist. The generalizability of these findings to non-math domains remains unclear. Math problems may have unique properties (e.g., objective ground truth, verification tractability) that make PDR effective in ways that don't transfer.
>
> **Response**: In the table below, we evaluate PDR on two additional benchmarks to assess generalizability beyond math. We use GPQA (diamond set), which contains graduate-level multiple-choice questions in biology, chemistry, and physics, and LiveCodeBench (2025 split), which consists of competitive programming problems from LeetCode, AtCoder, and Codeforces. LiveCodeBench results are reported with $B_{seq} = 65536$ tokens and GPQA results with $B_{seq} = 32768$ tokens. All numbers are reported as mean@16. Across both benchmarks, PDR consistently improves over our best non-PDR decoding baseline, suggesting that its benefits extend beyond math to other domains as well.
>
> | | LiveCodeBench | LiveCodeBench | GPQA (diamond) | GPQA (diamond) |
> | :--- | :---: | :---: | :---: | :---: |
> | | Long CoT | PDR ($w=[8], k=4$) | Long CoT | PDR ($w=[8], k=4$) |
> | **gemini-2.5-flash** | 55.96 | **61.25** | 75.98 | **82.32** |
> | **gpt-o3-mini** | 48.90 | **58.61** | 73.30 | **76.26** |
>
>
> > The distillation operator D is critical as it compresses multi-draft solutions into a bounded workspace. Further, the paper explores four heuristic distillation strategies (global summary, top-k, random-k). However, it does not provide principled criteria for when each strategy should apply, and explain why global summary and per-sample top-k succeed whereas random-k underperforms.
>
> **Response**: Our empirical results suggest two simple criteria for effective distillation: (i) retain as much high-quality signal as possible and (ii) preserve diversity across candidate solutions under a fixed token budget. Global summary and per-sample top-$k$ both satisfy these criteria better than random-$k$. Global summary first exposes the model to all drafts and then compresses them, so the distilled workspace tends to include the key reasoning steps from multiple distinct solution paths while removing redundancy. Per-sample top-$k$ uses the verifier to discard clearly low-quality drafts while retaining several diverse, high-scoring candidates per instance, which leads to stronger in-context guidance and more diverse follow-up generations than a single global top-$k$.
>
> Random-$k$, in contrast, ignores both quality and coverage: it often drops the few correct drafts and keeps unproductive ones, which explains its underperformance. In practice, when token and compute budgets permit, we recommend global summary or per-sample top-$k$. Random-$k$ is only preferable in extremely constrained regimes where additional LLM calls are expensive.
>
>
> > Evaluation on commercial APIs (gemini-2.5-flash, o3-mini) limits reproducibility and ablations. The authors cannot, for example, vary model size systematically or inspect model internals. Further, an ablation across model sizes is not shown. Typically, 8B dense model trained with operator RL is relatively small by modern standards. There should have been an evaluation on larger models, say 20B+ to show effectiveness of the approach.
>
>
> **Response**: We agree that relying only on commercial API models limits reproducibility. To address this, we now include additional experiments with the open-weight Qwen3 family of models in the table below.
>
> | | AIME 2025 | AIME 2025 |
> | :--- | :---: | :---: |
> | | Long CoT | PDR ($w=[8, 8], k=4$)
> | Qwen3-4B-Instruct-2507 | 43.13 | 56.67 |
> | Qwen3-30B-A3B-Instruct-2507 | 51.04 | 69.17 |
>
> We are currently running further Qwen3 experiments at additional model sizes and will update the paper with the complete size ablation.
>
> In addition, we report results on LiveCodeBench (2025-01–2025-06) and GPQA (diamond) for the GPT-OSS model family. We omit AIME for this series because scores are already saturated ($\geq$ 90% accuracy).
>
> | | LiveCodeBench | LiveCodeBench | GPQA (diamond) | GPQA (diamond) |
> | :--- | :---: | :---: | :---: | :---: |
> | | Long CoT | PDR ($w=[8], k=4$) | Long CoT | PDR ($w=[8], k=4$) |
> | GPT-OSS-20B | 55.62 | 63.99 | 61.55 | 68.50 |
> | GPT-OSS-120B | 63.12 | 69.71 | 69.92 | 72.29 |
>
> Together, these open-model experiments complement the API-based results and provide evidence that the effectiveness of PDR is not restricted to a single 8B model or to proprietary APIs.

---

> ### Author Response · Authors · 2025-11-25
> **Response to Reviewer 79KF (Part 2)**
>
> > Did you try an ablation on the mixing ratio? Try 25-75, 40-60, 60-40 splits in Equation 11. Does the 50-50 choice matter?
>
> **Response**: We did not perform an explicit ablation over the mixing ratio in Eq. (11), and use a 50–50 split as a simple default. Our goal was to jointly train two capabilities: (i) single-shot problem solving, and (ii) PDR-style generate → distill → refine, without introducing an additional tuned hyper-parameter.
>
> Intuitively, increasing the PDR weight should bias the model towards producing higher-quality summaries and refinements. Conversely, increasing the weight on the standard RL objective should favor single-turn performance when PDR is not used at inference time.
>
> > Can you describe scenarios where PDR underperforms: (1) Very short thinking budgets (do summaries compress too aggressively)? (2) Tasks requiring deep state tracking (does round-wise forgetting hurt)?
>
> **Response**: We did a manual inspection of some samples where PDR underperforms or matches standard CoT and found two interesting cases:
>
> - Few correct drafts among the N parallel samples: In these instances, only a small fraction of drafts are correct. Both top-$k$ and global summary then mix a large number of incorrect solutions with a tiny number of correct ones, without explicitly isolating the failure modes of the incorrect generations. Therefore, the summary does not reliably surface signals correlated with correctness. In such rare cases, a long and unconstrained CoT trajectory can sometimes follow the single correct reasoning path and outperform PDR, which is effectively averaging over mostly wrong trajectories.
>
> - No correct drafts among the N parallel samples: When all drafts are wrong, both long CoT and PDR predict incorrect answers. In principle, PDR should still be able to extract useful structure (partial progress, contradictions, etc.) but instead it over-weights a wrong pattern in subsequent rounds thus leading to summary drift (subsequent rounds refine and reinforce an incorrect line of reasoning) and wrong generations.
>
> More broadly, PDR’s gains are contingent on the model’s verification and refinement abilities: when these are weak, the distillation step can amplify incorrect patterns instead of correcting them, so PDR may not improve over standard CoT.
>
> > Figure 3 and Figure 9 show PDR dominates Long CoT on B_seq but SR dominates on B_total. This is stated but not deeply analyzed: (1) When would practitioners prefer SR (high latency tolerance, cost-constrained) vs. PDR? (2) How do parallelism constraints on typical hardware affect the tradeoff?
>
>
> **Response**: When the primary constraint is total token usage or overall compute (e.g., serving many users or running large offline evaluations), SR is preferred. For the same $B_{total}​$, SR allocates all tokens to a single trajectory, and our curves show it outperforms PDR under that constraint.
>
> When wall-clock latency per query is the main bottleneck and total compute is less constrained, PDR is more beneficial. PDR uses multiple shorter trajectories and a small number of refinement calls. With sufficient parallelism, the wall-clock latency is governed by $B_{seq}$ and PDR achieves higher accuracy than a single long SR/CoT trace.
>
> > How does performance change with different workspace sizes?
>
>
> **Response**: Due to compute constraints, we kept the memory workspace size fixed across our main experiments and did not run a full sweep over workspace sizes. Conceptually, we expect a trade-off. Increasing the workspace size allows PDR to retain more information from past iterations, which should improve performance up to a point. Beyond that point, we expect diminishing or even negative returns: as context grows very large, current LLMs' long-context abilities degrade. In practice, these models are pre-trained with a sequence context of 128K tokens and then extended via RoPE-based extrapolation during post-training to 1M+ tokens, but using the full extended context is rarely optimal; performance typically starts to drop beyond some fraction of the maximum window.

---

> > ### Comment · Reviewer_79KF · 2025-11-27
> > **Official Comment by reviewer 79KF**
> >
> > Thanks for the clarification. I will maintain my score.

---

> > > ### Author Response · Authors · 2025-11-27
> > > **Request for Feedback on Remaining Concerns**
> > >
> > > Thank you again for the detailed review and for taking the time to consider our responses.
> > >
> > > For our own understanding and to improve the paper (either in a camera-ready version or future work), could you let us know which of your original concerns you still view as the main limitations after our additional experiments and clarifications?
> > >
> > > Even a brief pointer to the most important remaining issues would be very helpful for us.

---

### Author Response · Authors · 2025-12-03
**Summary of review comments and rebuttal**

We sincerely thank all the reviewers for their feedback and highlighting the strengths of this work including iterative improvement using PDR/SR over Long CoT, PDR operator-consistent RL to reduce the train-test mismatch, pareto-frontier analysis of the various inference strategies for matched sequential and total token budgets, and impressive results on AIME 2024/2025.

We addressed all the questions/weaknesses during our rebuttal and we also thank reviewer 7aGP for mentioning that they'll increase their score. We provide a summary of the new experiments and clarifications below:

> Evaluation scope beyond AIME and math benchmarks

**Response**: We evaluate PDR on two additional benchmarks to assess generalizability beyond math. We use GPQA (diamond set), which contains graduate-level multiple-choice questions in biology, chemistry, and physics, and LiveCodeBench (2025 split), which consists of competitive programming problems from LeetCode, AtCoder, and Codeforces.

| | LiveCodeBench | LiveCodeBench | GPQA (diamond) | GPQA (diamond) |
| :--- | :---: | :---: | :---: | :---: |
| | Long CoT | PDR ($w=[8], k=4$) | Long CoT | PDR ($w=[8], k=4$) |
| gemini-2.5-flash | 55.96 | **61.25** | 75.98 | **82.32** |
| gpt-o3-mini | 48.90 | **58.61** | 73.30 | **76.26** |

Across both benchmarks, PDR consistently improves over our best non-PDR decoding baseline, suggesting that its benefits extend beyond math to other domains as well.

> Evaluation on open models like Qwen / GPT-OSS

**Response**: Two reviewers noted that evaluation on gemini-2.5-flash / gpt-o3-mini limits reproducibility. We added additional model results including GPT-OSS and the Qwen series of models and find similar consistent gains using PDR over Long CoT.

| | AIME 2025 | AIME 2025 |
| :--- | :---: | :---: |
| | Long CoT | PDR ($w=[8, 8], k=4$)
| Qwen3-4B-Instruct-2507 | 43.13 | 56.67 |
| Qwen3-30B-A3B-Instruct-2507 | 51.04 | 69.17 |

| | LiveCodeBench | LiveCodeBench | GPQA (diamond) | GPQA (diamond) |
| :--- | :---: | :---: | :---: | :---: |
| | Long CoT | PDR ($w=[8], k=4$) | Long CoT | PDR ($w=[8], k=4$) |
| GPT-OSS-20B | 55.62 | **63.99** | 61.55 | **68.50** |
| GPT-OSS-120B | 63.12 | **69.71** | 69.92 | **72.29** |

> How does PDR compare to self-consistency or best-of-N sampling?

**Response**: We add self-consistency results in the table below for both gemini-2.5-flash and gpt-o3-mini, and find PDR outperforms the self-consistency baseline as well.

| | AIME 2024 | AIME 2024 | AIME 2024 | GPQA (diamond) | GPQA (diamond) | GPQA (diamond) |
| :--- | :---: | :---: | :---: | :---: | :---: | :---: |
| | Long CoT | $maj@16$ (Self Consistency) | PDR ($w=[8], k=4$) | Long CoT | $maj@16$ (Self-consistency) | PDR ($w=[8], k=4$) |
| gemini-2.5-flash | 81.2 | 86.67 | 88.5 | 75.98 | 79.80 | 82.32 |
| gpt-o3-mini | 76.25 | 83.33 | 86.7 | 73.30 | 74.24 | 76.26 |

> PDR vs SR vs long CoT: when to use what

**Response**:

- At fixed total compute $B_{\text{total}}$ (cost/throughput constrained), SR is often preferred: it allocates most tokens to a single trajectory (or a small number), and our curves show it is very competitive under tight total budgets.
- At fixed sequential budget $B_{\text{seq}}$ (latency-constrained), PDR is more attractive: multiple short drafts plus summarization can be run in parallel, achieving higher accuracy at roughly the same per-query latency as a single long CoT/SR trace.

> Computational cost and time cost of PDR compared with Long CoT.

**Response**: We do a simple inference latency profiling test using an 8B model on 1 H100 GPU and achieve the following latencies:

| Configuration | Generation Budget | #LLM Calls | $B_{seq}$ | $B_{total}$ | Latency (s) |
| :---: | :---: | :---: | :---: | :---: | :---: |
| Long CoT | 16384 | 1 | 16384 | 16384 | 202.7761 |
| Long CoT | 73728 | 1 | 73728 | 73728 | 938.6954 |
| PDR ($w=[8], k=4$) | 8192 | 2 | 16384 | 73728 | 215.2382 |

PDR (row 3) is processing more total tokens compared to Long CoT (row 1), but we are able to leverage GPU parallelization to keep similar latencies and much higher accuracy for PDR compared to Long CoT for a fixed $B_{seq}$. If we consider $B_{total}$, PDR (row 3) latency is much lower and we still get better performance in PDR because the Long CoT (row 2) performance stagnates after a certain generation budget.

There's a tradeoff between compute spent (FLOPs) and accuracy, and PDR is able to extract out more performance by spending a lot more compute. The affect on latency is minor because of GPU parallelization. Additionally, this latency effect can be reduced further by deploying multiple model copies and reducing the batch size, which will decrease the minor latency gap between Long CoT and PDR while retaining the high performance of PDR compared to Long CoT.

More details on the computational cost and latencies are present in response to Reviewer 7aGP.

---

### Meta-Review · Area_Chair_poHa · 2026-01-07

**Summary:**

This paper reframes LLM reasoning as an iterative improvement process rather than a single long chain-of-thought, introducing Sequential Refinement (SR) and Parallel-Distill-Refine (PDR) with a bounded workspace, along with operator-consistent RL to align training and inference. Reviewers broadly find the framing intuitive and practically relevant, and the budget-aware evaluation (separating sequential vs. total token budgets) is a strong methodological contribution. Initial concerns focused on narrow evaluation scope (math-only), unclear guidance on when PDR vs. SR should be preferred, limited baseline coverage, and reproducibility due to reliance on closed models. The rebuttal substantially strengthens the paper by adding non-math benchmarks (GPQA, LiveCodeBench), open-model evaluations (Qwen, GPT-OSS), self-consistency comparisons, and a concrete latency profiling study, addressing most major objections. While some conceptual and cost-analysis limitations remain, the revised submission presents a compelling and well-supported case for short-context iterative reasoning as a practical alternative to long CoT.

**Reviewer Concerns:**

Addressed concerns -

* Added evaluations on GPQA and LiveCodeBench showing consistent gains for PDR.
* Extended experiments to open models (Qwen, GPT-OSS), mitigating reliance on closed APIs.
* Comparison to self-consistency / best-of-N

Concerns remained unaddressed -

* Although latency is analyzed, total FLOPs / energy cost and cost–benefit trade-offs remain only partially quantified.
* While global summary and top-k empirically work best, there is still no principled criterion predicting which strategy suits which task.
* Limited analysis of scenarios where PDR/SR underperform.

**Reviewer Scores:**

No

---

### Decision · Program_Chairs · 2026-01-26

Reject